# MIRROR TRAINING FOR INPUT CONVEX NEURAL NETWORK

## ABSTRACT

The input convex neural network (ICNN) aims to learn a convex function from the input to the output by using non-decreasing convex activation functions and non-negativity constraints on the weight parameters of some layers. However, in practice, it loses some representation power because of these non-negativity parameters of the hidden units, even though the design of the "passthrough" layer can partially address this problem. To solve issues caused by these non-negativity constraints, we use a duplication input pair trick, i.e., the negation of the original input as part of the new input in our structure. This new method will preserve the convexity of the function from the original input to the output and tackle the representation problem in training. Additionally, we design a mirror unit to address this problem further, making the network Mirror ICNN. Moreover, we propose a recurrent input convex neural network (RICNN) structure to deal with the time-series problems. The recurrent unit of the structure can be ICNN or any other convex variant of ICNN. This structure can maintain convexity by constraining the mapping from the hidden output at time step $t$ to the input of the next time step $t + 1$. The experiments can support our design, including the simple numerical curve fitting, power system hosting capacity dataset regression, and the MNIST dataset classification.

## 1 INTRODUCTION

Convex optimization's mathematical foundations have been researched for centuries, yet numerous recent developments have sparked new interest in this topic Hindi (2004). In machine learning, convexity and optimization typically refer to the optimization of the parameters or the minimization of the loss Bengio et al. (2005). However, the input convex neural network (ICNN) in Amos et al. (2017) provides a different perspective on the convexity of the neural network, which is from the input to the output.

The input convexity of the ICNN is preserved because of the non-decreasing convex activation function, such as the rectified linear unit (ReLU) Nair & Hinton (2010); Agarap (2018), and the non-negativity constraint on some of the hidden layers. These non-negative weights can maintain the convexity from the input to the output but also brings the problem of the lack of representation power. Though using the "passthrough" layers can partially provide substantial representation, this problem is still a challenge during the input convex neural network training.

To tackle the challenge in representation, Chen et al. (2018) concatenates the negation of the original input with itself, making it the new input of the network. This method can theoretically get more representation power because of the duplication input pair, but the actual training process does not work as expected. The reason for the bad convergence is the new non-negativity constraint on the "passthrough" layers. Therefore, we proposed the modified trick, mirror training, to improve the convergence of the training. We prove that the non-negativity constraint on the "passthrough" layers will lead to a poor training result. Moreover, the new designs in our model, the negative pair of input and the mirror unit, will not break the convexity from the input to the output and improve the training performance.

Because of the great convexity property of the ICNN, it has been widely used for different tasks. Chen et al. (2020) introduces the application of voltage regulation through the ICNN. In addition,

Chen et al. (2018) uses the ICNN for the building control task under both the single and time-series scenarios. Similarly, Kolter & Manek (2019) also extends the scope of the ICNN for dynamic models. Dynamic models or sequence models are always topics that require attention. To take advantage of the convexity of the ICNN and the sequence models, such as the RNN, we construct a loop using the ICNN as the recurrent unit, which is the recurrent ICNN. The network is convex from the sequential input to output by adding the non-negativity constraint on the weights of the hidden output. Our model is more straightforward to modify than the other sequence structure because the recurrent unit can be any convex variant of the basic ICNN.

The remainder of this paper is organized as follows. Section 2 discusses the related works. Section 3 illustrates the design of the mirror training and the structure of the recurrent input convex neural network. Section 4 provides the numerical validation using different datasets. Conclusion and discussions are in Section 5.

## 2 RELATED WORK

**Input convex neural network.** The input convex neural network (ICNN) is proposed in Amos et al. (2017). Given a fully connected neural network, the model can learn a convex function from the input to the output because of the non-negativity constraint on the weights of some hidden layers. However, this constraint reduces the representation power of the model even though the use of the "passthrough" layers can provide some additional representation. The solution provided by Chen et al. (2018) aims to address this problem by using the duplication input pair, but the new constraint on the "passthrough" layers will cause a problem in convergence. We provide the mirror training technique to tackle the problem and prove that the new structure can preserve convexity and have additional representation power.

**Recurrent neural network.** The recurrent neural network (RNN) is a class of sequence models that are widely used for various tasks, including time-series prediction Qin et al. (2017), machine translation Cho et al. (2014) and speech recognition Shewalkar (2019). The core design of the basic RNN is the recurrent unit, also known as an RNN "cell," where the output is connected to the input, forming a cycle. Many variants of the basic RNN achieve state-of-the-art performance by modifying the cell structure. For example, the long short-term memory (LSTM) Hochreiter & Schmidhuber (1997) and the gated recurrent unit (GRU) Chung et al. (2014) use the concept of "gate" to forget, select, and memorize the information flowing in the model, therefore learning the time-series relationship. Our design of the recurrent input convex neural network (RICNN) takes advantage of this recurrent structure. It formulates the cell as a basic ICNN to capture the convexity for time-series tasks. Chen et al. (2018) also proposed an input convex sequence model. The differences between this model and our network are that, first, the recurrent unit of our model can be any form of a convex network, while the cell of the model in Chen et al. (2018) only has one layer of full connected neural; second, our recurrent network does not need to make all weights non-negative.

**Hosting capacity analysis in the power system.** Hosting capacity analysis is a popular topic in power system research. The analysis determines how many more distributed energy resources (DERs) the power grid can host without causing technical issues Wu et al. (2022). Traditional hosting capacity analysis can be treated as an optimization problem Yuan et al. (2022). Nazir & Almassalkhi (2019) presents the hosting capacity analysis as a convex inner approximation of the optimal power flow problem. The data-driven hosting capacity analysis method is always formulated as a time-series problem to observe the hosting capacity value changes over time Rylander et al. (2018). In this paper, we use the proposed recurrent input convex neural network to consider the convexity of this analysis, meanwhile capturing the temporal correlation of the hosting capacity values.

## 3 METHOD

This section will explain how to obtain a convex mapping from the input to the output through the input convex neural network (ICNN). Moreover, we design a mirror training technique to increase

the representation power of the basic ICNN. The new structure is the Mirror ICNN. With the modified ICNN, we expand the work to a recurrent structure by proving that the new structure is still convex in terms of the input for the time-series tasks.

## 3.1 INPUT CONVEX NEURAL NETWORKS

The basic input convex neural network in Amos et al. (2017) is developed from the fully connected neural network. Given a fully connected $k$-layer neural network, we can re-construct it as an input convex neural network shown in Fig. 1a. The mathematical expression of the network is

$$\boldsymbol{z}_1 = \sigma_0(\boldsymbol{U}_0\boldsymbol{x} + \boldsymbol{b}_0), \tag{1}$$

$$\boldsymbol{z}_{i+1} = \sigma_i(\boldsymbol{W}_i\boldsymbol{z}_i + \boldsymbol{U}_i\boldsymbol{x} + \boldsymbol{b}_i), \tag{2}$$

where $h_i$ denotes the output of the $i$-th hidden layer in the neural network, $\boldsymbol{W}_{1:k}$ and $\boldsymbol{U}_{0:k}$ are the parameters of the fully connected layers and the "passthrough" layers, respectively, and $g_i$ is the activation functions.

The convexity from the input $\boldsymbol{x}$ to the output $\boldsymbol{y}$ is achieved following Proposition 1.

**Proposition 1** *The neural network is convex from the input to the output, given that all weights in* $\boldsymbol{W}_{1:k-1}$ *are non-negative, and all of the activation functions* $\sigma(\cdot)$ *are convex and non-decreasing.*

The proof of Proposition 1 follows the operations that preserve convexity mentioned in Boyd & Vandenberghe (2004). First, a non-negative weighted sum of convex functions is convex. Second, for a function composition, i.e., $f(x) = h(g(x))$, $f$ is convex if $h$ is convex and non-decreasing, and $g$ is convex. Therefore, the ICNN $f(x)$ is convex with respect to input $x$. By splitting the input features into different parts, i.e., $\boldsymbol{x}$ and $\boldsymbol{o}$, and only adding the non-negativity constraint on $\boldsymbol{x}$ related weights, the partially input convex neural network (PICNN), as shown in Fig. 1b, can learn a function $f(\boldsymbol{x}, \boldsymbol{o})$ which is convex with respect to $\boldsymbol{x}$.

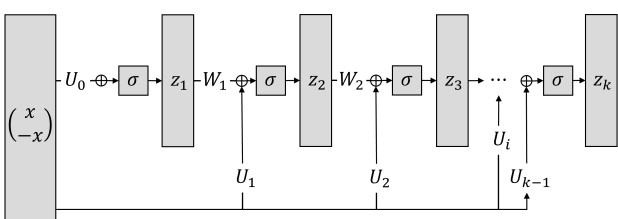

(a) The structure of the fully input convex neural network.

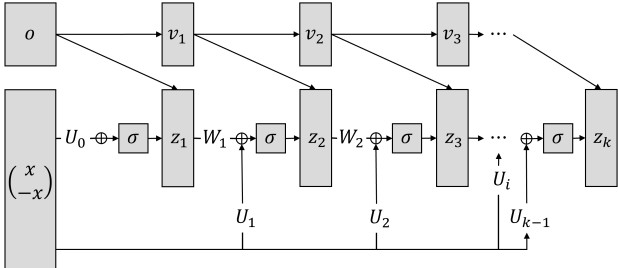

(b) The structure of the partially input convex neural network.

Figure 1: The input convex neural network.

The requirement of the non-decreasing convex activation functions is not restricted. We can choose from some popular activation functions, e.g., rectified linear unit (ReLU) and exponential linear unit (ELU), which are already proven to be powerful in learning. However, the non-negativity constraint on $\boldsymbol{W}_{1:k-1}$ will limit the representation power, leading to inadequate training. The "passthrough" layers are designed to address this problem, but the linear mapping cannot compensate for all the lost representation power. In this case, we use mirror training to tackle this drawback fundamentally.

## 3.2 MIRROR TRAINING FOR ICNN

Because of the non-negativity constraint on the weights, the ICNN loses much representation power, even though the linear mapping of the "passthrough" layer is designed to address this issue. To fundamentally solve this problem, we propose using a mirror training trick for the ICNN, which contains the duplication input pair and the mirror training unit for hidden layers.

First, we will introduce the duplication input pair. Besides the original input $\boldsymbol{x}$, we also use the negation of the input, i.e., $-\boldsymbol{x}$, as part of the input. The new structure with the duplication input pair, $\begin{bmatrix} \boldsymbol{x} \\ -\boldsymbol{x} \end{bmatrix}$, is shown in Fig. 1a. The mathematical equations can be written as

$$\boldsymbol{z}_1 = \sigma_0(\boldsymbol{U}_0 \begin{bmatrix} \boldsymbol{x} \\ -\boldsymbol{x} \end{bmatrix} + \boldsymbol{b}_0), \tag{3}$$

$$\boldsymbol{z}_{i+1} = \sigma_i(\boldsymbol{W}_i \boldsymbol{z}_i + \boldsymbol{U}_i \begin{bmatrix} \boldsymbol{x} \\ -\boldsymbol{x} \end{bmatrix} + \boldsymbol{b}_i). \tag{4}$$

Following Proposition 2, the new input pair will not impact the input convexity.

**Proposition 2** *If* $f(\begin{bmatrix} \boldsymbol{x} \\ -\boldsymbol{x} \end{bmatrix})$ *is a convex function,* $g([\boldsymbol{x}]) = f(\begin{bmatrix} \boldsymbol{x} \\ -\boldsymbol{x} \end{bmatrix})$ *is also a convex function, i.e., the function* $f$ *is also convex with respect to* $\boldsymbol{x}$.

The proof of Proposition 2 is simple. Boyd & Vandenberghe (2004) shows that the composition with an affine mapping will preserve the convexity. If $f$ is convex, $g(\boldsymbol{x}) = f(\boldsymbol{A}\boldsymbol{x} + \boldsymbol{b})$ is also convex. Suppose the ICNN learns a convex function $g(\begin{bmatrix} \boldsymbol{x} \\ -\boldsymbol{x} \end{bmatrix}) : \mathbb{R}^{2n} \to \mathbb{R}$ and $f(\boldsymbol{x}) : \mathbb{R}^n \to \mathbb{R}$. Using matrix $\boldsymbol{A}$ which maps $\begin{bmatrix} \boldsymbol{x} \\ -\boldsymbol{x} \end{bmatrix}$ to $[\boldsymbol{x}]$, we have Proposition 2.

Chen et al. (2018) also uses a similar negation duplicate pair to retrieve the representation power of the basic ICNN. Differently, in the "passthrough" layers $\boldsymbol{U}_{0:k-1}$ of the model in Chen et al. (2018), the negative weights in $\boldsymbol{U}_{0:k-1}$ of the basic ICNN structure are set to zero, and their negations are set as the weights for corresponding $-\boldsymbol{x}$. This way, the forward calculation in the new structure has the same result as the calculation in the original ICNN structure. For example, suppose we have an input pair $[x_1, x_2]^T$ and the corresponding weights pair is $[w_1, w_2]$, where $w_2 < 0$. Following the statement, we can get a new weights pair $[w_1, 0, 0, -w_2]^T$. With the new input pair $[x_1, x_2, -x_1, -x_2]$, the result is $(w_1 x_1 + w_2 x_2)$, which equals the inner product of $[w_1, w_2]$ and $[x_1, x_2]$.

However, in practice, this method proposed in Chen et al. (2018) will not work as expected. First, the network will reset all weights of $-\boldsymbol{x}$ after each iteration, so these weights $\boldsymbol{U}_{0:k-1}^{(-)}$ are only considered in the forward calculation, not in the backpropagation. Second, since all the weights of $\boldsymbol{x}$, i.e., $\boldsymbol{U}_{0:k-1}^{(+)}$, are non-negative after each iteration, the negation of their negative value, which is the weights in $\boldsymbol{U}_{0:k-1}^{(-)}$ will be small in the new iteration and to the final iteration. Third and most important, the "passthrough" layers are linear mapping of the input, so we can quickly know that the $\boldsymbol{U}_{0:k-1}$ with no constraint are equivalent to any other linear transformation in the network, including the negation trick of $\boldsymbol{U}_{0:k-1}^{(+)}$ and $\boldsymbol{U}_{0:k-1}^{(-)}$.

By proving that we do not need the non-negativity constraint for $\boldsymbol{U}_{0:k-1}$, Proposition 2 is enough to preserve the convexity from the input to the output.

Second, for the mirror training unit shown in Fig. 2, denoting $\tilde{\boldsymbol{x}} = \begin{bmatrix} \boldsymbol{x} \\ -\boldsymbol{x} \end{bmatrix}$, we have the equation as

$$\boldsymbol{z}_{i+1} = \sigma_i(\boldsymbol{W}_i \boldsymbol{z}_i + \boldsymbol{U}_i \tilde{\boldsymbol{x}} + \boldsymbol{b}_i), \tag{5}$$
$$\boldsymbol{z}'_{i+1} = \sigma_i(\boldsymbol{M}_i \boldsymbol{z}_i - \boldsymbol{U}_i \tilde{\boldsymbol{x}} + \boldsymbol{b}'_i). \tag{6}$$

The convexity of the $\boldsymbol{z}'_{i+1}$ can be achieved following Proposition 3. This structure of $z_{i+1}$ and $z'_{i+1}$ will use the output of the "passthrough" layers in a coupled way. Because of the introduced duplication input pair of $\tilde{\boldsymbol{x}}$ and the mirror training unit, the ICNN can fundamentally gain representation power. We name the new structure a mirror input convex neural network (Mirror ICNN or MICNN).

**Proposition 3** $z_{i+1}$ *is convex, given that all weights in* $\boldsymbol{M}_i$ *are non-negative, and all of the activation functions* $\sigma(\cdot)$ *are convex and non-decreasing.*

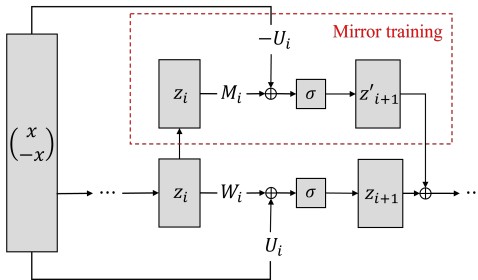

Figure 2: The mirror training unit for the input convex neural network.

### 3.3 RECURRENT INPUT CONVEX NEURAL NETWORK

By the design of mirror training, the input convex neural network can have a convex mapping from the input to the output. However, the ICNN is not naturally designed for sequential tasks, but there are many of them in practice. For example, dynamic hosting capacity analysis is a popular research topic in the power system area. To capture the convexity of a time-series problem, we extend the ICNN to a time-series form. In this case, we design a recurrent structure for the ICNN, i.e., the recurrent input convex neural network (Recurrent ICNN or RICNN).

Like the basic recurrent neural networks (RNNs), we construct our model using the recurrent unit, the "cell." Different from the cells in the RNN network, the recurrent unit in our RICNN network is an ICNN network. Denoting the output of the ICNN at sequence $(t)$ as $\boldsymbol{y}^{(t)}$, we have the RICNN model shown in Fig. 3. The mathematical definition of the model can be expressed as

$$\boldsymbol{z}_1^{(t+1)} = \sigma_0(\boldsymbol{U}_0\tilde{\boldsymbol{x}}^{(t+1)} + \boldsymbol{H}_0\boldsymbol{y}^{(t)} + \boldsymbol{b}_0), \tag{7}$$

$$\boldsymbol{z}_{i+1}^{(t+1)} = \sigma_i(\boldsymbol{W}_i\boldsymbol{z}_i + \boldsymbol{U}_i\tilde{\boldsymbol{x}}^{(t+1)} + \boldsymbol{H}_i\boldsymbol{y}^{(t)} + \boldsymbol{b}_i), \tag{8}$$

where the hidden state $\boldsymbol{y}^{(t)}$ is part of the input at sequence $(t+1)$ and $\boldsymbol{H}_{0:k-1}$ are the corresponding matrix in the "passthrough" layers. The convexity from the input $\boldsymbol{x}$ to the output $\boldsymbol{y}$ of the RICNN network is maintained following Proposition 4.

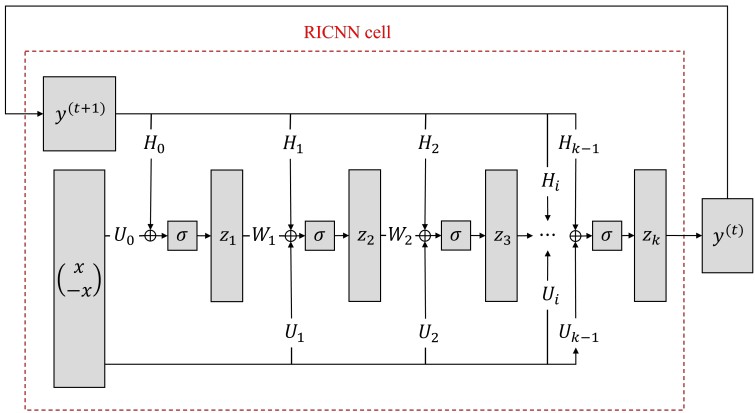

Figure 3: The recurrent input convex neural network.

**Proposition 4** *The proposed recurrent neural network is convex from the input to the output, given that all weights in* $\boldsymbol{W}_{1:k-1}$ *and* $\boldsymbol{H}_{0:k-1}$ *are non-negative, and all of the activation functions* $\sigma(\cdot)$ *are convex and non-decreasing.*

The proof of Proposition 4 is similar to Proposition 1. Note that $\boldsymbol{y}^{(t)}$ in this network is a convex function of $\boldsymbol{x}^{(t)}$. Therefore, the matrix $\boldsymbol{H}_{0:k-1}$ should be non-negative to maintain the convexity from $\boldsymbol{x}$ to $\boldsymbol{y}$. The recurrent unit in the RICNN can be replaced by any other convex neural network, which will not break the convexity from the input to the output.

## 4 EXPERIMENTS

This section will provide numerical validation on the design of the mirror training trick and the recurrent input convex neural network.

### 4.1 FITTING CONVEX CURVES

The input convex neural network (ICNN) can learn a convex function from the input to the output. To visualize this basic property, we design a fitting curve experiment to deliver a direct perception of the ICNN. Given a function $f(x)$, we use basic ICNN and proposed mirror training for ICNN to this function. The results are shown in Fig. 4. Both the ICNN and the Mirror ICNN can perform well for easier tasks, such as $f(x) = x^2$ and $f(x) = e^{x^2}$. However, the ICNN shows its drawback when facing a harder task with a relatively large negative bias, i.e., $f(x) = e^{x^2} - 3000$, where the proposed mirror training for ICNN can overcome the challenge.

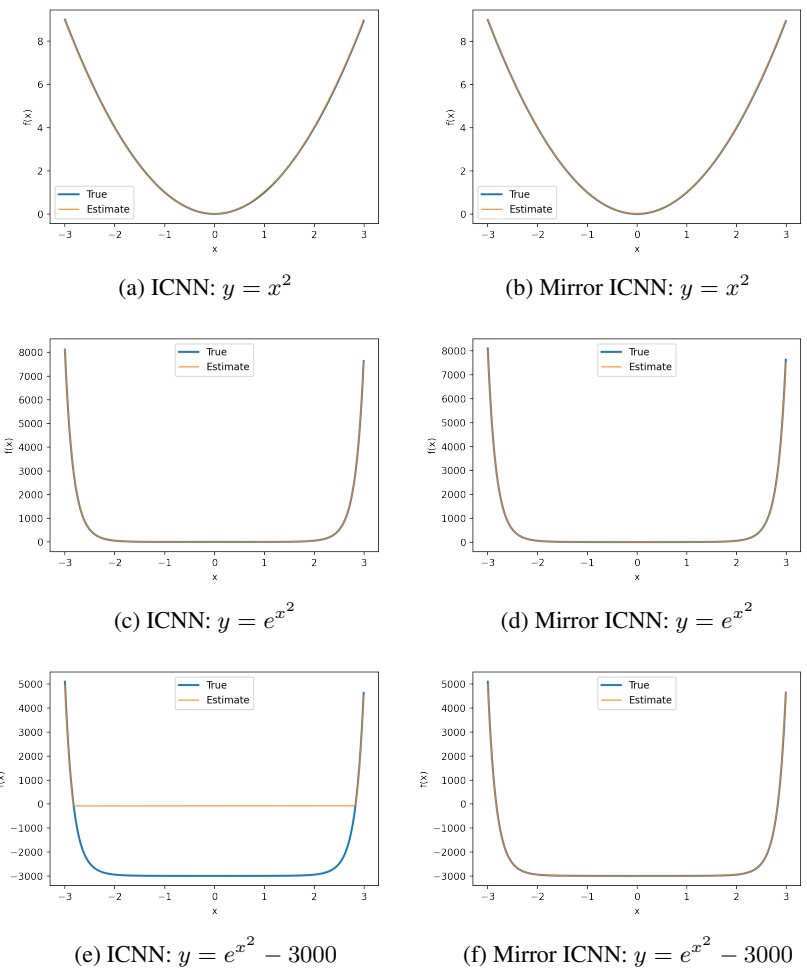

(a) ICNN: $y = x^2$      (b) Mirror ICNN: $y = x^2$

(c) ICNN: $y = e^{x^2}$      (d) Mirror ICNN: $y = e^{x^2}$

(e) ICNN: $y = e^{x^2} - 3000$      (f) Mirror ICNN: $y = e^{x^2} - 3000$

Figure 4: Training results of the fitting curve task using different models.

### 4.2 EXPERIMENTS ON THE HOSTING CAPACITY DATASET

The hosting capacity (HC) value is denoted as the maximum active power that can be injected by DERs at a bus in an existing distribution grid without causing technical problems or requiring changes to power system facilities. We use the IEEE 123-bus example feeder model as our physical model. The input data is the observation of the power system data of each bus, including the real power, the reactive power, etc. The output data is the hosting capacity value of the corresponding bus. The training results are shown in Fig. 5, while the mean square errors (MSE) in training are shown in Table 1.

We use three models to experiment; the first one is the basic ICNN proposed in Amos et al. (2017), the second one is the modified ICNN proposed in Chen et al. (2018), which introduces the new non-negativity constraint on the "passthrough" layers, and the third one is the structure proposed in this paper with the mirror training. The results show that our model can reach the smallest MSE in a smoother training process.

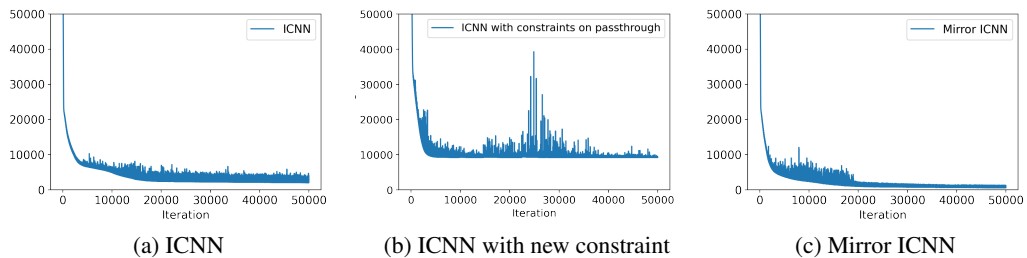

(a) ICNN        (b) ICNN with new constraint        (c) Mirror ICNN

Figure 5: Training results of the hosting capacity dataset using different models.

Table 1: Training MSE of the hosting capacity dataset using different models.

| Model | Training MSE |
|---|---|
| ICNN | 713.26 |
| ICNN with new constraint | 9222.87 |
| Mirror ICNN | **509.98** |

### 4.3 EXPERIMENTS ON MNIST DATASET

MNIST is a dataset of handwritten digits LeCun (1998), which is a classic classification task in machine learning. We use the 3-layer neural network as our baseline. The used ICNN and Mirror ICNN have the same quantity of hidden neurals as the baseline.

By observing the result, as shown in Table 2, they all have similar testing accuracy on this dataset. When zooming into the first 100 iterations of the training, we can see that the Mirror ICNN has both faster and better performance, as shown in Fig. 6.

Table 2: Testing accuracy of MNIST dataset using different models.

| Model | Testing accuracy |
|---|---|
| NN | $97.70\% \pm 0.06\%$ |
| ICNN | $97.45\% \pm 0.06\%$ |
| Mirror ICNN | $\mathbf{97.96\% \pm 0.07\%}$ |

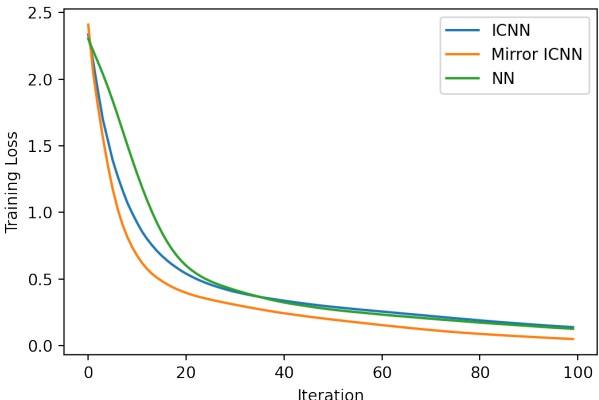

Figure 6: The first 100 iterations of the training using different models.

## 4.4 RECURRENT PREDICTION ON THE HOSTING CAPACITY DATASET

To evaluate the performance of the recurrent input convex neural network, we design an experiment on the hosting capacity dataset. Currently, many researchs show the value of dynamic hosting capacity analysis, i.e., hosting capacity number change over time. The IEEE 123-bus example feeder model is our physical model. The input data is the time-series observation of the power system data of each bus. The output data is the hosting capacity value of the corresponding bus.

We choose the LSTM model as our baseline. As shown in Table 3, the mean square losses on both models are very close. However, the Recurrent ICNN model has a faster convergence speed, as shown in Fig. 7.

Table 3: Training results of time-series hosting capacity dataset using different sequence models.

| Model | Training MSE |
|-------|--------------|
| LSTM | 1358.09 |
| RICNN | 1327.65 |

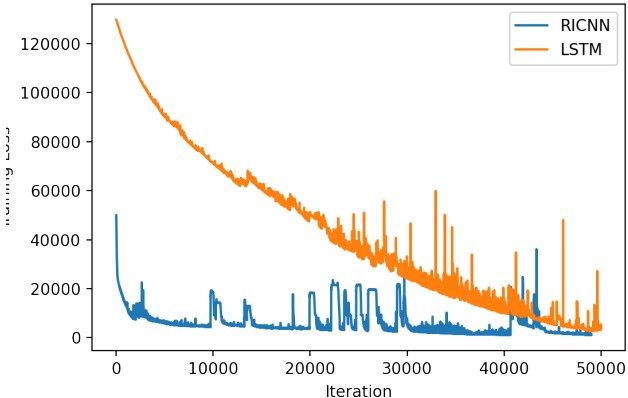

Figure 7: The first 50000 iterations of the training using different models.

# 5 CONCLUSION

The input convex neural network (ICNN) learns a convex function from the input to the output. However, because of the non-negativity constraint on the hidden layers, it loses some representation power. This paper proposes a mirror training technique to address this problem for the ICNN. This mirror training, namely mirror input convex neural network (Mirror ICNN or MICNN), includes the duplication input pair and the mirror training unit for the ICNN. The results of different datasets show that this technique will improve training performance. Moreover, to extend the ICNN to time-series tasks, we design the recurrent input convex neural network (Recurrent ICNN or RICNN). The recurrent unit, which can be any convex variant of the basic ICNN, takes the output of the last time as part of the input.

During the training, we found that though the proposed new structure of the mirror unit will have a better performance, the training cost is relatively high compared to the other model. In our future work, we plan to find a strategy to reduce the training cost. Additionally, we believe the learned convex function can be composed with other structures as the physical enhancement of the learning. Therefore, we plan to explore more application scenarios for the input convex neural network.

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
