# OpenReview forum: "Mirror Training for Input Convex Neural Network"
_ICLR.cc/2023/Conference — Submitted to ICLR 2023_

### Official Review · Reviewer_Ngme · 2022-10-24

**Confidence:** 4
**Correctness:** 3
**Technical Novelty And Significance:** 3
**Empirical Novelty And Significance:** 2
**Recommendation:** 5

**Clarity, Quality, Novelty And Reproducibility:**

The paper is clear and it is easy to understand the idea. However, the experiments need more elaborations, which is a crucial problem. The current experiments cannot demonstrate the advantages of the proposed mirror ICNN.

**Details Of Ethics Concerns:**

The paper does not have ethics concerns.

**Strength And Weaknesses:**

Strength:
The paper is well-written. The idea is simple and clear. The proposed mirror training with a duplication input pair is very practical to address the representation problem of ICNN.

Weaknesses:
The claims are not well-supported by the current experiments. The detailed comments are below.
1.	The mirror ICNN uses a duplication input pair. So its weights U_i and W_i are larger than those of the basic ICNN. What is a fair comparison between ICNN and mirror ICNN? It needs to give more detailed descriptions on the issue. At the end of the first paragraph in subsection 4.3, it claims that it uses the same quantity of hidden neurons. What does it mean?
2.	It is better to include ICNN, modified ICNN by Chen et al. 2018, and the proposed mirror ICNN in all the comparisons. For example, it needs to compare with the modified ICNN by Chen et al. 2018 in Figure 4.
3.	The compared neural network are all shallow. It is better to test with deep networks.
4.	Which activation function is used in the experiment? Does the used activation function affect the results?
5.	In Figure 5 and Table 1, why the modified ICNN proposed by Chen et al. (2018) performs so poor? It is known that the modified ICNN gets more representation power than the basic ICNN.
6.	There are four propositions in the paper. It should distinguish between what propositions proposed in this paper and what propositions in others.
7.	It is better to use three-line tables.

**Summary Of The Paper:**

The paper considers the representation problem of the input convex neural network (ICNN). It uses a duplication input pair trick from the work of Chen et al. (2018). Notability, the paper presents a mirror unit and a mirror training trick for ICNN. It also extends a recurrent structure for the ICNN to handle time-series problems. Experiments test the effectiveness of the proposed method.

**Summary Of The Review:**

The input convex neural network (ICNN) has the representation problem because of these non-negativity parameters of the hidden units. The paper presents the mirror training trick with the duplication input pair to address this problem. The idea is simple and clear. However, the experiments need more elaborations, which is the main weakness.

---

### Official Review · Reviewer_72S5 · 2022-10-24

**Confidence:** 4
**Correctness:** 2
**Technical Novelty And Significance:** 2
**Empirical Novelty And Significance:** 2
**Recommendation:** 3

**Clarity, Quality, Novelty And Reproducibility:**

# Clarity

The paper and contributions are clearly articulated. I have a question on the power of the mirror terms that is not clear from the text. Section 3.2 states:

> Because of the non-negativity constraint on the weights, the ICNN loses much representation power,
even though the linear mapping of the “passthrough” layer is designed to address this issue. To
fundamentally solve this problem, we propose using a mirror training trick for the ICNN, which
contains the duplication input pair and the mirror training unit for hidden layers.

A significant limitation of convolutional ICNNs is that many of the convolutional layers are constrained to have non-negative weights, which prevent them from being able to capture edge filters, for example, that require positive and negative values in each filter. Is my understanding correct that the mirror ICNN does **not** help with the issue of many convolutional layers not being able to have positive and negative values in every kernel?

# Quality and novelty

Where there could be a nice modeling contribution in this paper, I do not think the experimental quality is good enough to validate the contribution.

# Reproducibility

There is no reproducibility statement and no source code or additional experimental details are provided

**Strength And Weaknesses:**

Strengths
+ The non-negativity constraints of the ICNN is harmful to the representational capacity and this paper tries to improve upon this
+ The focus on improving power systems is a strong application area with large potential to improve

Weaknesses
+ The main weakness is that experimental results are unfortunately not well-contextualized and make it difficult to understand where the real contribution is coming from. There are many of established experimental settings using ICNNs (cited within the paper), but the paper does not explicitly quantitatively build or improve upon any of them. Because the contribution of the paper is meant to be a new model that should improve ICNNs, it is important to show how the model performs in the many existing settings.
+ The hosting capacity task in Section 4.2 seems similar to the results in [Section VI of Chen et al. 2020](https://arxiv.org/pdf/2002.08684.pdf) as they both use the IEEE 123-bus test feeder, but this connection is not mentioned. None of the experimental results refer back to the results originally published in Chen et al. 2020. For example, it would be extremely convincing to take Table 1 from Chen et al. 2020 and add a "Mirror ICNN" column that improves upon their results.

**Summary Of The Paper:**

This paper proposes an input-convex neural network (ICNN) variant that considers negating the input to increase the capacity (Section 3.2/Figure 2) as well as a recurrent variant (Section 3.3/Figure 3). Section 4 experimentally compares these methods for fitting synthetic curve fitting tasks (Fig 4), a power system hosting capacity task (Section 4.2/4.4), and MNIST classification (Section 4.3).

**Summary Of The Review:**

I recommend to reject the paper because the new ICNN variant being proposed is not evaluated on any established settings using ICNNs, which make the contribution difficult to understand.

---

### Official Review · Reviewer_p9y3 · 2022-11-03

**Confidence:** 4
**Correctness:** 4
**Technical Novelty And Significance:** 2
**Empirical Novelty And Significance:** 2
**Recommendation:** 5

**Clarity, Quality, Novelty And Reproducibility:**

The paper is clear and well written, no doubt about that. My main concern is with the superficial/minor nature of the result.

From what I understand the paper essentially consists in augmenting the original inputs with their negation and using the result as an input to each of the layers. Although there is some inventiveness in the duplicating the input in order to compensate for the non negative weights, as indicated by the authors themselves, several of the ideas were already present in previous work (see for example Chen et al. 2018-2020 for the use of mirrored inputs including both the original inputs and their negation, Amos et al. for the notion of input convex neural networks and Chen et al. 2018 for the introduction of input convex recurrent neural networks). The paper does not strongly differentiate itself from all those works.





**Strength And Weaknesses:**

The core material of the paper appears relatively light/incremental (at this stage) with respect to the results in Amos et al. and Chen et al. (As an illustration of this, compare Figures 1 and 2 in Amos et al.  to Fig. 1 in the present paper)

The work is still at an early stage (although there are some interesting foundations such as Fig. 4(e) and (f)). The exposition gives an impression of dilution and lacks substance. As a few examples of this, a whole page is used to discuss previous work, another page is used to give intuition on the proof of proposition 1 which can be found in Amos et al. 2017.

The paper lacks a clear (theoretical or even more elaborate numerical) comparison between the expressive power of mirror networks and simple input convex networks (or networks with the mirroring step restricted to the first layer). I.e It would be great if you could provide theoretical guarantees for a class of functions that cannot be captured by vanilla input convex neural networks but which can be expressed by your new mirrored architecture.



**Summary Of The Paper:**

The authors introduce and study a novel architecture for input convex neural networks (i.e. neural networks that are designed to be convex as a function of the inputs). The traditional architecture of ICNNs is improved through the addition of a "mirroring" step, appending to the original input, the negation of this input and using the resulting concatenation as an input to each of the layers (which is slightly different from previous work where the concatenation was used solely as an input to the first layer). The paper proves convexity of the resulting architecture and then goes on to use this architecture in the setting of recurrent neural networks.

**Summary Of The Review:**


General comments
- I understand why convexity w.r.t. the weights is useful but I don’t understand why convexity w.r.t the input is interesting

Detailed comments

page 1
- I would remove the proof of proposition 1 (honestly you can just keep one line (if you really want) recalling that the composition of convex functions is still a convex function and that non negative combination maintains convexity)

page 4
- Same comment as for proposition 1. You can drop the proof and just add a link to Boyd and Vandenberghe? In fact I would remove the proof of proposition 2 and expand on the paragraph “However, in practice, this method proposed in Chen et al. (2018) will not work as expected ” which is unclear (see my comments below)
- At the end of the page, you write “By proving that we do not need the non-negativity constraint for U0:k−1, Proposition 2 is enough to preserve the convexity from the input to the output”.
- The notation z_i’ is unclear. the expression of both z_{i+1} and z_{i+1}’ depend on the previous z_i but z_{i+1} seems to never be reused. This is also unclear in your Figure 2 as this figure only represents one step

page 5
- In (8) when you write z_i, I guess you mean z_i^{(t+1)}? and you do not define y^{(t)} which from the diagram should be given as z_{K}^{(t)}. I would recommend adding this to (7) and (8)
- In your diagram (Fig 3) for the recurrent neural network, I think the y^{(t+1)} should be changed into a y^{(t)}. I.e. I find it confusing that your y^{(t)} suddenly becomes a y^{(t+1)} in the feedback

page
- Fig. 4. You only compare the prediction from your architecture with the simple ICNN. How about the ICNN in Chen et al where the mirroring step is already used (albeit only in the first layer)?

page 7
- “The used ICNN and Mirror ICNN have the same quantity of hidden neurals “ —> do you mean “hiden neurons” ?

page 8
- For the hosting capacity dataset, it would be interesting to understand the difference between the minimizers returned by the training of the LSTM and RICNN networks. Are those minimizers comparable in terms of generalization? if not, how do they differ?

Typos:
- “The requirement of the non-decreasing convex activation functions is not restricted” —> “is not restrictive”
- on page 2, after the proof of proposition 2, “uses a similar negation duplicate pair” —> perhaps what you mean is “use a duplicate of x” ? or “add the nagation of the original inputs”?
- again after proposition 2, I would replace “Differently, ” by “on the contrary” or “however”
- page 8, first paragraph of section 4.4. “Currently, many researchs show the value” —> “many/several results”

---

### Official Review · Reviewer_m7TB · 2022-11-04

**Confidence:** 4
**Correctness:** 2
**Technical Novelty And Significance:** 1
**Empirical Novelty And Significance:** 1
**Recommendation:** 1

**Clarity, Quality, Novelty And Reproducibility:**

The paper has major technical flaws; the major statement for method motivation and contribution is incorrect and the experimental evaluation is flawed and fails to adequately support the main claims.

**Strength And Weaknesses:**

Strength:

The topic of studying input convex neural networks is interesting and applies to many real-world applications.

Weakness:

1. The paper is poorly written and many statements are wrong. For example, One of the big motivations for this paper as mentioned in the Introduction part of the paper is "These non-negative weights can maintain the convexity from the input to the output but also brings the problem of the lack of representation power... To tackle the challenge in representation, Chen et al. (2018) concatenate the negation of the original input with itself, making it the new input of the network. This method can theoretically get more representation power because of the duplication input pair, but the actual training process does not work as expected. " <-- this statement is wrong.

The original input convex neural network paper [Amos et al. 2017] and the modified ICNN structure paper [Chen et al. 2018] both have the representation power to represent all convex functions. As mentioned by [Chen et al. 2018] page 4, "our construction of ICNN in Proposition 1 is motivated by [12] but modified to be more suitable to control of dynamical systems... Our construction achieves the exact representation as [12]..." [Chen et al. 2018] introduced the duplication trick because they not only want the function to be convex, but they need the composition of ICNN functions to be still convex - which is important for dynamical system modeling where the network needs to be “rolled out in time”.

If the objective of ICNN is not for dynamical system control - like in all the experiments shown in this paper, from fitting convex curves, fitting the hosting capacity, and MNIST. I don't see a need for not using the original ICNN structure in [Amos et al. 2017]; also the proposed structure does not seem to provide any better representation-wise or training-wise benefits compared to [Amos et al. 2017].

2. The Experiment section feels pretty sloppy and rushed. All experiments are toy-scale problems not meeting the bar of a top AI conference.

3. The experiment results are also mysterious and not convincing. For example, For the MNIST experiment 4.3, why NN has a worse performance than ICNN? NN has much better representation power and a simple search on Google will give you a trained NN that can obtain 99% accuracy in the test set. Also, if you look at Fig 6, does the training even converge? Same for Figure 7, the LSTM curve - the training loss is still fast decreasing.

4. Also, the major content in the Method parts are from previous papers, e.g., Proposition 1 is from [Amos et al. 2017], and the negation method is from [Chen et al. 2018]. The technical contribution of the paper seems quite marginal.

**Summary Of The Paper:**

This paper proposes a modified input convex neural network structure and demonstrated the performance of the proposed structure in learning single-input single-output convex functions, hosting capacity dataset, and MNIST.

**Summary Of The Review:**

This paper proposes a modified ICNN structure, with empirical results in learning convex functions, power systems and MNIST.

---

### Decision · Program_Chairs · 2023-01-20

**Decision:**

Reject

**Justification For Why Not Higher Score:**

 The benefits of the proposed architecture are not clear, and authors did not address the concerns in a rebuttal.

**Justification For Why Not Lower Score:**

 NA

**Metareview: Summary, Strengths And Weaknesses:**

Thank you for your submission to ICLR.  The reviewers and I are in agreement that while there are some interesting ideas, the proposed paper is not yet ready for publication.  In particular, the proposed work is based upon an alternative structure of ICNNs that avoid non-negativity of the hidden units; however, this formulation is not sufficiently justified in theory or in practice, and it is thus unclear what benefits the methods provide over the original or updated ICNN formulations.  As the authors didn't respond to these critiques, it seems reasonable that the paper should not be accepted.